# Towards Efficient and Diverse Generative Model for Unconditional Human Motion Synthesis

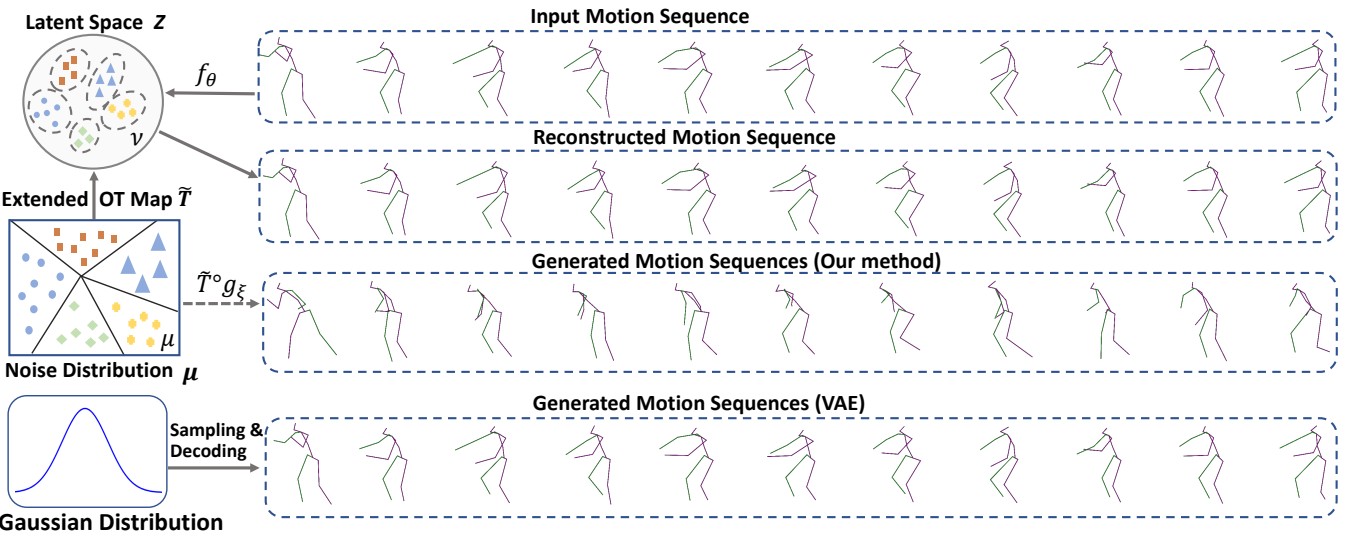

**Figure 1: The comparison synthesized human motion sequences based on the end pose of samples. The vanilla VAE randomly samples latent codes from a prior distribution which are then decoded into human motions that only reside in the major patterns of the human motions. The proposed MOOT aims to unconditionally synthesize high-quality and diverse human motions by combining the extended optimal transport (OT) map and generator.**

## ABSTRACT

Recent generative methods have revolutionized the way of human motion synthesis, such as Variational Autoencoders (VAEs), Generative Adversarial Networks (GANs), and Denoising Diffusion Probabilistic Models (DMs). These methods have gained significant attention in human motion fields. However, there are still challenges in unconditionally generating highly diverse human motions from a given distribution. To enhance the diversity of synthesized human motions, previous methods usually employ deep neural networks (DNNs) to train a transport map that transforms Gaussian noise distribution into real human motion distribution. According to Figalli's regularity theory, the optimal transport map computed by DNNs frequently exhibits discontinuities. This is due to the inherent limitation of DNNs in representing only continuous maps. Consequently, the generated human motions tend to heavily concentrate on densely populated regions of the data distribution, resulting in mode collapse or mode mixture. To address the issues, we propose an efficient method called MOOT for unconditional human motion synthesis. First, we utilize a reconstruction network based on GRU and transformer to map human motions to latent space. Next, we employ convex optimization to match the noise distribution with the latent space distribution of human motions through the Optimal Transport (OT) map. Then, we combine the extended OT map with the generator of reconstruction network to generate new human motions. Thereby overcoming the issues of mode collapse and mode mixture. MOOT generates a latent code distribution that is well-behaved and highly structured, providing a strong motion prior for various applications in the field of human motion. Through qualitative and quantitative experiments, MOOT achieves state-of-the-art results surpassing the latest methods, validating its superiority in unconditional human motion generation.

**Unpublished working draft. Not for distribution.**

## CCS CONCEPTS

• **Applied computing** → *Media arts*.

## KEYWORDS

Unconditional human motion synthesis, generative model, optimal matching

# 1 INTRODUCTION

Human motion synthesis task aims to generate diverse and coherent human motion sequences that satisfy specific cues or spatio-temporal constraints. This field has wide-ranging applications, such as human motion understanding [31, 39], autonomous driving [19, 20, 41, 42], and animation [12, 13]. Unconditionally generating diverse human motions from a given data distribution still presents a significant challenge, especially when the human motion datasets are diverse, unstructured, and unlabeled. Although current human motion generation methods have shown promising ability to produce human motions, fully capturing the potential diversity from a given distribution for unconditional human motion synthesis remains a challenge.

Recent advancements in deep generative models, including Variational Autoencoders (VAEs) [6, 18, 21, 35], Generative Adversarial Networks (GANs) [8, 17, 23], and Denoising Diffusion Probabilistic Models (DMs) [1, 32], emerge as the dominant approaches for capturing the data distribution cross all possible human motions. Specifically, VAEs leverage an encoder network to map the data distribution into a Gaussian latent distribution. This representation of latent space is mapped back to the data distribution using a decoder network. Although significant progress has been made, VAE-based methods assume Gaussian distribution for the latent space of the data distribution. This assumption often leads to the generation of blurry human motion joints, thereby reducing the accuracy of the generated human motions. As shown in the bottom of Figure. 1, we visualize the results of the end pose of different motion sequences, most human motions using VAE tend to converge to the same pattern. Another popular approaches are the GAN-based methods, an unconditional generator is trained to generate real-like motions from random noise, while a discriminator is employed to measure the distinction between the generated samples and real ones. Training GANs can be challenging and sensitive to hyperparameters. Moreover, GANs tend to sample human motions from the major patterns while ignoring the minor patterns. DMs-based methods define a diffusion process, in which Gaussian noise is gradually added to the input data using a Markov chain. As DMs-based methods rely on multiple iterations of denoising diffusion steps, each requiring a substantial amount of computation, the overall inference time can be prohibitively long for large-scale datasets or complex models.

The purpose of previous human motion generative methods is to calculate a transport map using DNNs. This transport map transforms Gaussian white noise into the distribution of human motions. By pushing the initial noise distribution forward, the transport map generates a new distribution that approximates real human motions. The similarity between these two distributions determines the generalization ability of the designed method [3]. As pointed out by previous works [22, 33], DNNs can only represent continuous mappings. The transport map may be discontinuous when the human motion data has multiple modes [22]. This phenomenon would incur the *mode collapse* or *mode mixture*, in which the generated human motion samples heavily concentrate around the most dense areas of the dataset while neglecting other less frequent or sparse regions of the data distribution during sampling. The sampled human motions often have less diversity and look unrealistic. The problem still exists even when the data distribution has a single

mode. These phenomena show that the mode collapse or mode mixture cannot be resolved in existing human motion generative methods.

To overcome the limitations, we propose a novel method called MOOT, for unconditional human motion synthesis that leverages optimal transport mapping. Our motivation is to address the mode collapse problem and mode mixture, thereby enhancing the diversity while ensuring the accuracy of generated human motion sequences. Previous generative methods for human motion generation mainly consist of two steps, i.e., manifold embedding and probability distribution transport. Manifold embedding aims to find the encoding and decoding maps between the data manifold embedded in the human motions and the latent space. Probability distribution transport aims to transport a given noise distribution to the data distribution. In this work, the proposed MOOT separates the process into two stages to enhance the diversity of unconditional human motion synthesis. Specifically, in the first stage, a human motion reconstruction network is designed to learn the manifold embedding of human motions. The manifold embedding is implemented by Gated Recurrent Units (GRU) and transformer [30], which aims to capture temporal smooth and spatial relations between human motions. For the second stage, we employ the optimal transport (OT) mapping to achieve the probability distribution transport and generate diverse human motions. According to Brenier's theorem [5], the optimal transport map can be denoted by the gradient map of the Brenier potential, which is continuous in the computing process. The OT mapping intends to map the noise distribution into the latent space of human motions, which can avoid the mode collapse and mode mixture, thereby enhancing the diversity of generated human motions. Figure. 1 briefly describe the human motion synthesis process of the proposed method and vanilla VAE. The proposed MOOT can generate more diverse human motion sequences.

In summary, our main contributions are as follows:

- We propose a novel method called MOOT for unconditional human motion synthesis, which aims to avoid mode collapse and mode mixture based on the theory of optimal transportation mapping.
- We employ the encoder of reconstruction network, based on GRU and transformer, to encode human motions into latent space. Then we utilize the Brenier potential between the noise distribution and latent space of human motions to represent the optimal transport map. The extended OT map combines the generator to generate new human motions.
- The proposed method is evaluated on widely-used human motion datasets in the comprehensive experiments. The obtained results demonstrate the effectiveness of the proposed method over the state-of-the-art approaches for unconditional human motion generation task.

# 2 RELATED WORK

## 2.1 Human Motion Generation

The emergence of neural networks has a significant impact on the field of human motion synthesis. Previously, the focus was primarily on specific human motion generation, conditioned on some limiting factors, such as music [16], text [38], and action

[25]. Numerous research papers have explored these areas. These works share a common spirit with our research, and here we will elaborate on them further. For example, Li et al. [16] proposed the task of generating a long sequence of realistic 3D dance motions that are well correlated with the input music. Petrovich et al. [25] learned an action-conditioned latent representation by training a VAE network. They sample from the learned latent space and query a series of positional encodings to synthesize motion sequences conditioned on actions. Degardin et al. [7] fused the architectures of GANs and GCNs to synthesize the kinetics of human motions. Zhang et al. [38] incorporated Denoising Diffusion Probabilistic Models (DDPM) [10] into motion generation, and propose to guide the generation pipeline with input texts softly, which increases the diversity of the generation results.

Another line of work focuses on unconditional human motion synthesis [26]. Holden et al. [11] presented a pioneer work in deep unconditional human motion synthesis. They propose a motion synthesis and editing method based on a deep learning framework, which can automatically learn an embedding of human motion in a non-linear manifold. Recent MDM [32] utilized a classifier-free diffusion-based generative model for the human motion domain, and reported a trading-off between diversity and fidelity of human motions under an unconditional human motion generation. Modi [26] employs the style of StyleGAN to synthesize human motions, they utilize a mapping network to map noise into this latent space to enhance the diversity.

Previous generative methods for human motion generation map the noise Gaussian distribution to the latent space distribution through a deep neural network. These methods usually suffer from mode collapse or mode mixture. In this work, we propose a new perspective human motion generation method to enhance diversity of the generated human motion sequences.

## 2.2 Generative Models

Given the inherent indeterminacy of future human motions, the generative models like variational autoencoders (VAEs) [34, 36, 40], generative adversarial networks (GANs) [2, 23] and denoising diffusion probabilistic model (DDPM) [1, 29] are appropriate methods for diverse human motion prediction due to their capability of generating a diverse set of valid solutions. Generally, VAEs [34] utilize the encoder-decoder mechanism to handle the task by capturing the probability distribution of human motions. VAEs-based methods assume that the latent distribution of human motions follows a Gaussian mixture distribution. This assumption often leads to blurry human motion joints, resulting in poor performance in terms of detail and accuracy in the generated human motions. GANs [8] can promote the quality of generated human motions in a degree. While being a powerful tool in generating realistically looking samples, GANs suffer from the mode collapse problem. The major modes (with higher likelihood) corresponding to a particular human motion pattern will more likely generate samples, minor patterns (those with lower likelihood) will almost not generate any samples. Recent promising diffusion model [10] has been proposed as an alternative generative approach for human motion generation task. For example, Tevet et al. [32] designed a Motion Diffusion Model (MDM) for various human motion generation, enabling different

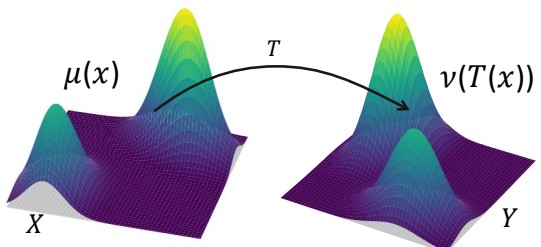

**Figure 2: Given two probability distributions $\mu$, $\nu$, and cost functions $c$, the Monge formalism is to find a optimal transport map $T$ that minimizes the cost functions. The transport map $T$ transforms the source $\mu$ into $\nu$.**

modes of conditioning, and different generation tasks. Ye et al. [37] attempted to introduce physics simulation, which is based on a pre-trained physical model through reinforcement learning. However, their approach is computationally intensive and low-efficiency inference process.

Recent studies by [33] realized that the mode collapse and mode mixture are caused by the discontinuous functions with continuous DNNs. To alleviate this, Nagarajan et al. [22] proposed to utilize gradient-based regularization to address these issues. Khayatkhoei et al. [15] propose the utilization of multiple GANs to alleviate mode collapse and mode mixture. Another approach that has shown success in overcoming mode collapse is the normalized diversification method. Pan et al. [24] proposed UniGAN with a Normalizing Flow-based generator and uniformity regularization. Song et al. [28] utilized a new training objective that additionally optimized over the generated samples. However, if the target data distribution has multiple modes, the transport map of previous generative works is discontinuous. DNNs can only represent continuous mappings, the intrinsic conflict causes mode collapse. In this work, we propose to employ optimal transport to avoid mode collapse for unconditional human motion synthesis.

## 3 PRELIMINARIES

**Optimal Transport Problem** Let $\mu \sim \mathcal{P}(X)$ and $\nu \sim \mathcal{P}(Y)$ be two sets of probability measures defined on $X$ and $Y$. $X \subset \mathbb{R}^d$ and $Y \subset \mathbb{R}^d$ are two subset of $d$-dimensional Euclidean space $\mathbb{R}^d$. The density functions are: $\mu(x) = f(x)dx$, $\nu(y) = g(y)dy$. The total measures are equal, $\mu(x) = \nu(y)$. The optimal transport problem can be attributed to Monge's problem, as shown in Figure. 2. Given a cost function $c(x, y): X \times Y \to [0, +\infty]$, which indicates the cost of moving each unit mass from the source $x \in X$ to the target $y \in Y$, the Monge's problem seeks the most efficient $\mu$-measurable map $T: X \to Y$ by

$$(MP) \quad \inf_T \int_X c(x, T(x))d\mu(x) \qquad (1)$$
$$\text{subject to} \quad \nu = T_\# \mu,$$

where $T_\# \mu$ denotes the push-forward measure induced by $T$. The map $T$ is measure-preserving: if for any measurable set $B \sim Y$, the set $T^{-1}(B)$ is $\mu$-measurable and $\mu(T^{-1}(B)) = \nu(B)$. The measure-preserving condition is denoted as $T_\# \mu = \nu$. A minimum $T^*$ to this problem is called an OT map. Intuitively, Monge's problem finds a

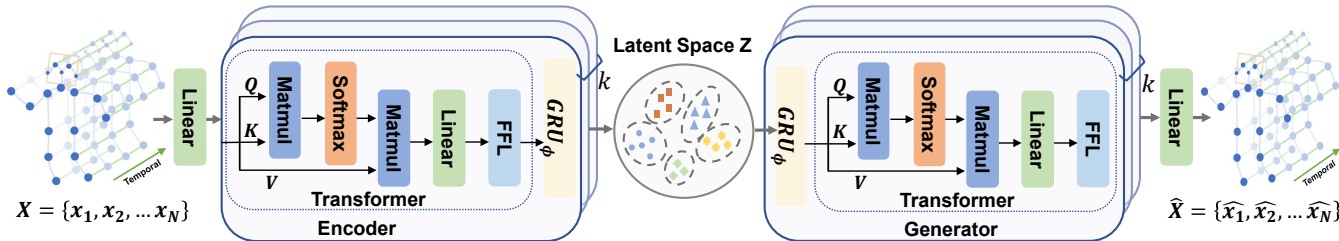

**Figure 3: The illustration of the human motion reconstruction network, which consists of the encoder and generator. Both are implemented by the GRU and transformer network.**

transport to turn the mass of $\mu$ into $\nu$ at the minimal cost measured by the cost function $c$.

**Optimal Transportation Map** The solution to Monge's problem is called the optimal transportation map, whose total transportation cost is called the Wasserstein distance between $\mu$ and $\nu$, denoted as $\mathcal{W}_c(\mu, \nu)$.

$$\mathcal{W}_c(\mu, \nu) = \min_{T_\#\mu=\nu} \int_X c(x, T(x))d\mu(x). \tag{2}$$

However, it has two drawbacks, $\mu$-mass cannot be split leading to hard constraints, its transport map may not exist. To overcome the above shortcomings, Kantorovich [15] relaxed transport maps into transport plans, and in turn, raised the Kantorovich problem.

**Kantorovich Problem** Suppose $X \subset \mathbb{R}^d$, $Y \subset \mathbb{R}^d$ are two subsets of the Eucidean space $\mathbb{R}^d$, $\mu = \sum_{i=1}^m \mu_i \delta(x - x_i)$, $\nu = \sum_{j=1}^n \nu_j \delta(y - y_j)$ are two discrete probability measure defined on $X$ and $Y$ with equal total measure, $\sum_{i=1}^m \mu_i = \sum_{j=1}^n \nu_j$. Then the Kantorovich Problem aims to find the optimal transport plan $P = (p_{ij})$ that minimizes the total transport cost:

$$(KP) \quad min_{P \in \pi(\mu, \nu)} \sum_{i=1}^m \sum_{j=1}^n p_{ij} c_{ij} \tag{3}$$

where $c_{ij} = c(x_i, y_j)$ is the cost that transports one unit mass from $x_i$ to $y_j$, and $\pi(\mu, \nu) = \{P| \sum_{i=1}^m p_{ij} = \nu_j, \sum_{j=1}^n p_{ij} = \mu_i, p_{ij} \geq 0\}$ is the coupling measure.

**Brenier's Approach** For quadratic Euclidean distance cost, the existence, uniqueness, and intrinsic structure of the optimal transportation map were proven by Brenier [5].

**Briener Potential Theorem** Suppose $X$ and $Y$ are the Euclidean space $\mathbb{R}^d$ and the transportation cost is the quadratic Euclidean distance $c(x, y) = 1/2\|x - y\|^2$. Furthermore $\mu$ is absolutely continuous, $\mu$ and $\nu$ have finite second order moments, $\int_X |x|^2 d\mu(x) + \int_Y |y|^2 d\nu(y) < \infty$, then there exists a convex function $u : X \to \mathbb{R}$, the so-called Briener potential, its gradient map $\nabla u$ gives the solution to the Monge's problem,

$$(\nabla u)_\#\mu = \nu. \tag{4}$$

The Brenier potential is unique upto a constant, hence the optimal transportation map is unique. $u$ is called a Brenier solution.

**Legendre Transform** Given a function $\varphi : \mathbb{R}^n \to \mathbb{R}$, its Legendre Transform is defined as follows:

$$\varphi^*(y) := \sup_x (\langle x, y \rangle - \varphi(x)). \tag{5}$$

## 4 METHOD

In this section, we first briefly introduce the overview of our task from an optimal transport perspective. Then we describe the reconstruction process of human motions. Finally, we explain how we compute the optimal transport map, including the matching learning and the process of generating new samples.

### 4.1 Overview

An overview of the MOOT method is illustrated in Figure. 1. Our key idea is to synthesize diverse and natural human motions unconditionally by training a human motion generation model. The MOOT mainly involved in two stages, one is the human motion reconstruction (Section 4.2), the other is optimal mapping and generation module (Section 4.3). The human motion reconstruction network, implemented by the GRU and transformer network, which intends to learn the manifold embedding of human motions, the detail is described in Figure. 3. The optimal mapping and generation module aims to calculate the optimal transportation map $T$ that aligns the noise distribution with the latent space distribution of human motions, then generate new human motions by combining the generator of reconstruction network, the detail is described in Figure. 4. Given a random noise, the proposed method can generate diverse and natural human motion sequences through the learned optimal transport mapping.

### 4.2 Human Motion Reconstruction

The human motion reconstruction network aims to learn the manifold embedding of human motions, the detail is described in Figure. 3. Specifically, given a sequence of human motion sequence $X = \{x_1, x_2, \cdots, x_N\}$, the encoder $f_\theta$ is trained to encode the human motions into latent space $Z$, which maps the human motion distribution into the latent code distribution. Then the generator $g_\xi$ decodes the latent representation to the data manifold. In this work, both the encoder and generator are implemented by the GRU and transformer network. As shown in Figure. 3, the transformer focuses on the inter-dependencies among human joints within the same time step. For the transformer process, when modeling the human motion sequences at time $t$ ($t \in \{1, 2, \ldots, T\}$), we project the whole sequence of joint embedding $E_t = [e_t^{(1)}, \ldots, e_t^{(a)}]^N$ into matrices $Q, K$, and $V$ by $W_Q, W_K, W_V$. $Q = E_t W_Q$, $K = E_t W_K$, and $V = E_t W_V$. The number of used human joint is $a$ ($a \in \{1, 2, \ldots, A\}$). The summary of the spatial joints $\tilde{E}_t$ is calculated by aggregating all the joint information using the multi-head mechanism. The $GRU_\phi$

**Figure 4: The process of computing optimal transport (OT) map $T$ and generating new samples. The extended OT map $\tilde{T}$ aligns with the latent space of human motions, and combines with the generator to sample new human motions. The red triangle $c_i$ denotes the center of each cell $W_i$.**

with parameter $\phi$ intends to capture the smoothness property of human motions, and then encode the human motions into latent space $Z$. The formula for using the encoder to map the human motions into latent space is computed as follows:

$$Attention(Q, K, V) = softmax(\frac{QK^T}{\sqrt{d_k}})V,$$

$$head_i = Attention(\boldsymbol{Q}^{(i)}, \boldsymbol{K}^{(i)}, \boldsymbol{V}^{(i)}), \qquad (6)$$

$$\tilde{E}_t = Concat(head_1, \ldots, head_H)\boldsymbol{W}^{(O)},$$

$$Z =\leftarrow GRU_\phi(\tilde{E}_t),$$

where $W^{(O)}$ denotes the concatenation weight matrix, the attention is computed by dot products of the query $\boldsymbol{Q}$ with all keys $\boldsymbol{K}$, divide each by $\sqrt{d_k}$, and apply a softmax function to obtain the weights on the values $\boldsymbol{V}$. In addition, the architecture of the generator is the same as the encoder. The generator aims to map the latent space $Z$ back to the reconstructed human motion sequence $\hat{X}$. To train the reconstruction network, the mean-square-error (MSE) loss function is utilized to measure the difference between the reconstructed and the original motion sequences. Specifically, we constrain the per-joint $L_2$ distance on human joints. $X$ and $\hat{X}$ are the input and the reconstructed motion sequences, respectively. The reconstruction loss is defined as follows:

$$L(X, \hat{X}) = \sum_{t=1}^{T} \sum_{a=1}^{A} \| x_t^{(a)} - \hat{x}_t^{(a)} \|_2 \qquad (7)$$

where $A$ and $T$ denote the number of joints and the generated sequence length, respectively. The calculated results contain the spatial and temporal information of human motion sequences, and map the human motions into latent space $Z$. The human motion reconstruction process is the first stage of MOOT, which learns the manifold embedding of human motions.

## 4.3 Optimal Mapping and Generation Module

After learning the process of human motion reconstruction, previous generative methods aimed to increase the diversity of human motions by utilizing a generator based on DNNs. This generator transformed a known continuous distribution, such as Gaussian white noise, into the real distribution of human motions. However, the sampled human motions from the noise tend to concentrate primarily on the most dense areas of the dataset, resulting in mode collapse and mixture. To address this issue, we introduce optimal

transport mapping to generate a wider range of diverse human motions, which consists of two stages, i.e., *optimal matching modeling* and *optimal sample generation*.

**Optimal Matching Modeling.** Optimal matching modeling aims to match the noise distribution with the latent space of human motions, which involves learning the optimal transport map (OT map) $T$, as shown in Figure. 4. In this work, the noise distribution $\mu$ is continuous that defined on a convex domain $\Omega \subset \mathbb{R}^d$, the latent space domain of human motions is a discrete set, $Z = \{z_1, z_2, \cdots, z_N\}, z_i \in \mathbb{R}^d$, which is a Dirac measure, $\nu = \sum_{i=1}^{n} \nu_i \delta(z - z_i), i = 1, 2, \ldots, N$, with the equal total mass as the noise measure, $\mu(\Omega) = \sum_{i=1}^{n} \nu_i$. We random draw $M$ samples from $\mu$ distribution, under a transport map $T : \Omega \rightarrow Z$, a cell decomposition is induced $\Omega = \bigcup_{i=1}^{n} W_i$, such that every $m$ in each cell $W_i$ is mapped to the target $z_i, T : m \in W_i \mapsto z_i$. The map $T$ is measure preserving, denoted as $T_\# \mu = \nu$, if the $\mu$-volume of each cell $W_i$ equals to the $\nu$-measure $T(W_i) = z_i, \mu(W_i) = \nu_i$. The cost function is given by $c : \Omega \times Z \rightarrow \mathbb{R}$ represent the cost for transporting a unit mass from $m$ to $z$. The total cost of $T$ is given by $\int_\Omega c(m, T(m))d\mu(m) = \sum_{i=1}^{n} \int_{W_i} c(m, z_i)d\mu(m)$. The introduced optimal transport map is the measure-preserving map that minimizes the total cost, $T^* := \arg\min_{T_\# \mu = \nu} \int_\Omega c(m, T(m))d\mu(m)$.

When the cost function is the $L^2$ distance $c(m, z) = 1/2\|m - z\|^2$, Brenier's theorem claims that the optimal transport map is given by the gradient map of a piece-wise (PL) convex function, the so-called Brenier potential $u_h : \Omega \rightarrow \mathbb{R}, u_h(m) := \max_{i=1}^{n} \{\pi_{h,i}(m)\}$, where $\pi_{h,i}(m) = \langle m, z_i \rangle + h_i$ is the hyperplane corresponding to $z_i \in Z$. The vector $h$ is the unique optimizer of the following convex energy under the condition that $\sum_i h_i = 0$,

$$E(h) = \int_0^h \sum_{i=1}^{n} w_i(\eta)d\eta_i - \sum_{i=1}^{n} h_i \nu_i, \qquad (8)$$

where $w_i(\eta)$ is the $\mu$-volume of $W_i(\eta)$. The convex energy $E(h)$ can be optimized simply by gradient descend method with $\nabla E(h) = (w_i(h) - \nu_i)^T$. The key is to compute the $\mu$-volume $w_i(h)$ of each cell $W_i(h)$. We draw $M$ random samples from noise distribution, we can find $W_i$ in which $m_j \in W_i$ by $i = \arg\max_i \{\langle m_j, z_i \rangle + h_i\}, i = 1, 2, \ldots, M$. When $M$ is large enough, $\hat{w}_i(h)$ converges to $w_i(h)$. The gradient of the energy is approximated as $\nabla E(h) \approx (\hat{w}_i(h) - \nu_i)^T$. We use adam algorithm to minimize the energy. Sampling of $m$ is independent of each other and finding the cell that $m$ is located only involves matrix multiplication and sorting. In this way, each

---

**Algorithm 1** Optimal Sample Generation.

---

**Require:** Latent codes $Z = \{z_1, z_2, \cdots, z_N\}$, latent code distribution $v$, number of random samples $M$, number of samples to generate $p$.

**Ensure:** Generated new samples $p$.

1: Initialize $E(h) = zeros(n)$.
2: **repeat**
3:     Sample $M$ uniformly from noise distribution $\{m_j\}_{j=1}^M$.
4:     Calculate $\nabla E(h) = (\hat{w}_i(h) - v_i)^T$.
5:     $\nabla E(h) = \nabla E(h)$ - $mean(\nabla E(h))$.
6:     Update $E(h)$ by adam algorithm.
7: **until** Converge
8: Optimal transport map $T(\cdot) \leftarrow \nabla(\max_i \langle \cdot, z_i \rangle + h_i)$.
9: **repeat**
10:     Sample $m \sim \mu$, Find the smallest $d + 1$ vertex around $m$ as $\{c_{i_0}, c_{i_1}, \ldots, c_{i_d}\}$.
11:     Generate latent code $\tilde{T}(m) = \sum_{k=0}^d \lambda_k z_{i_k}$ with $\sum_{k=0}^d \lambda_k = 1$.
12: **until** Generate new samples $p$.

---

random sample from the noise distribution is aligned with the latent code distribution.

**Optimal Sample Generation.** The above process describes the matching modeling process that aims to align latent space with the noise distribution, but this process does not generate new samples. Therefore, we extend the optimal transport map $T$ to a piecewise linear mapping $\tilde{T}$. The projection in the source domain induces a cell decomposition of $\Omega$, of which each cell is of $\mu$-volume $v_i$ and is mapped to the corresponding $z_i$. By representing the cells of $\mu$-volume centers as $c_i := \int_{W_i(h)} m d\mu(m)$, then the point-wise map $c_i \mapsto z_i$. The cell decomposition induces a triangulation of the centers $C = \{c_i\} : if W_i \cap W_j \neq \emptyset$, then $c_i$ is connected with $c_j$ to form an edge $[c_i, c_j]$. Similarly, if $W_{i_0} \cap W_{i_1} \cdots \cap W_{i_k} \neq \emptyset$ then there is a $k$-dimensional simplex $[c_{i_0}, c_{i_1}, \ldots, c_{i_k}]$. All these simplices form a triangulation of $C$ (a simplicial complex), denoted as $\mathcal{T}(C)$. We can triangulate $Z$ in the same way to obtain the triangulation $\mathcal{T}(Z)$. Once a random sample $m$ is drawn from the distribution $\mu$, we can find the simplex $\sigma$ in $T(C)$ containing $m$. Assume the simplex $\sigma$ has $d + 1$ vertices $\{c_{i_0}, c_{i_1}, \ldots, c_{i_d}\}$, the bary-centric coordinates of $m$ in $\sigma$ is defined as $m = \sum_{k=0}^d \lambda_k c_{i_k}$, and $\sum_{k=0}^d \lambda_k = 1$ with all $\lambda_k$ non-negative. Then the generated latent code of $m$ under this piece-wise linear map is given by $\tilde{T}(m) = \sum_{k=0}^d \lambda_k z_{i_k}$, then the $m$ is mapped to be $\tilde{T}(m)$. $z_i$ is used to construct the simplicial complex $\mathcal{T}(Z)$ in the support of the target distribution, we can guarantee that no mode is lost. The algorithm for optimal sample generation is shown in Algorithm. 1.

In summary, MOOT can synthesize diverse and natural human motion sequences through the above-mentioned two stages. The human motion reconstruction network learns a structured well-structured laent space of human motions through an encoder. The optimal matching aims to match the noise distribution with the latent space distributions. To generate new samples, we extend the OT map to point-wise mapping $\tilde{T}$, and then combining the generator of the reconstruction network to generate human motions. Finally, MOOT can fully capture the potential diversity from a given distribution for unconditional human motion synthesis.

# 5 EXPERIMENTAL DESIGN

In this section, the experimental design will be described, encompassing the popular human motion datasets, parameter settings, evaluation metrics, and baseline methods.

## 5.1 Datasets

The experiments are conducted on four widely used motion capture datasets, i.e., Human3.6M (H3.6M) [14], HumanEva-I [27], HumanAct12[9], and GRAB [4]. **H3.6M** consists of 11 subjects and 3.6 million frames at 50 Hz. The proposed method trains a model on five subjects (S1, S5, S6, S7, and S8) and is then tested on two subjects (S9 and S11). **HumanEva-I** contains 3 subjects recorded at 60 Hz, each of which performs 5 action categories. The pose is represented by 15 joints. **HumanAct12** contains 12 subjects in which 12 categories of actions with per-sequence annotation are provided. The sequences with less than 35 frames are removed, which results in 727 training and 197 testing sequences. Subjects P1 to P10 are used for training, P11 and P12 are used for testing. **GRAB** consists of 10 subjects interacting with 51 different objects, performing 29 different actions. We use 8 subjects (S1-S6, S9, S10) for training and the remaining 2 subjects (S7, S8) for testing.

## 5.2 Parameter Settings

For the parameter settings of the experiments, the batch size is set to 128, and the number of used human joints is 16. The number of multi-head in transformer is set to 8. The number of vertices $d$ to compute the bary-centric is set to 5. In this work, $\mu = v = \frac{1}{N}$. The sampled $M$ and $N$ are set to 4990 and $1000 \times 128$, respectively. The proposed method is implemented based on the PyTorch framework in Python 3.6. To guarantee the convergence of the proposed method, the Adam optimizer is adopted to train our model. The learning rate is initially set to $10^{-2}$ with a 0.98 decay every 10 epochs. The proposed method is trained for 500 epochs.

## 5.3 Evaluation Metrics

In addition, for a fair comparison, our method employs the following evaluation metrics: APD, ADE, FDE, Multi-Modal ADE, and FDE metrics (MMADE and MMFDE). APD metric aims to evaluate the diversity of the results, while the other four metric aim to evaluate the accuracy. In particular, lower is better for all metrics except the APD metric.

## 5.4 Baseline Methods

In this work, we compare the proposed method with state-of-the-art methods using the VAEs, GANs and DMs for unconditional human motion synthesis, including Dlow [36], MT-VAE [34], MOJO [40], HP-GAN [2], GSPS [21], BeLFusion [1], MotionDiff [32], MDM [29] and ACTOR [25].

# 6 RESULTS AND ANALYSIS

The section endeavors to provide a comprehensive analysis of the experimental results for accuracy and diversity, including the quantitative comparison results between the state-of-the-art methods and the proposed method, ablation studies, and qualitative analysis.

**Table 1: The comparison results between the proposed method and state-of-the-art methods on Human3.6M and HumanEva-I datasets. The best results are in bold. Lower is better for all metrics except the APD metric.**

| Method | Human3.6M | | | | | HumanEva-I | | | | |
|---|---|---|---|---|---|---|---|---|---|---|
| | APD ↑ | ADE ↓ | FDE ↓ | MMADE ↓ | MMFDE ↓ | APD ↑ | ADE ↓ | FDE ↓ | MMADE ↓ | MMFDE ↓ |
| DLow(ECCV'20) | 11.741 | 0.425 | 0.518 | 0.495 | 0.531 | 4.855 | 0.251 | 0.268 | 0.362 | 0.339 |
| MOJO(CVPR'21) | 12.579 | 0.412 | 0.514 | 0.497 | 0.538 | 4.181 | 0.234 | 0.244 | 0.369 | 0.347 |
| MT-VAE(ECCV'18) | 10.403 | 0.457 | 0.595 | 0.716 | 0.883 | 9.021 | 0.345 | 0.403 | 0.518 | 0.577 |
| MoDi(CVPR'23) | 17.57 | 0.761 | 0.635 | 0.818 | 0.799 | 12.730 | 0.402 | 0.534 | 0.497 | 0.538 |
| HP-GAN(CVPRW'18) | 7.214 | 0.858 | 0.867 | 0.847 | 0.858 | 1.139 | 0.772 | 0.749 | 0.776 | 0.769 |
| GSPS(ICCV'21) | 14.757 | 0.389 | 0.496 | 0.476 | 0.525 | 5.825 | 0.233 | 0.244 | 0.343 | 0.331 |
| MDM(arXiv'22) | 16.024 | 0.602 | 0.616 | 0.714 | 0.721 | 15.126 | 0.345 | 0.403 | 0.518 | 0.577 |
| Actor(ICCV'21) | 14.104 | 0.625 | 0.810 | 0.532 | 0.354 | 13.239 | 0.334 | 0.244 | 0.369 | 0.347 |
| BeLFusion(ICCV'23) | 7.602 | 0.372 | 0.474 | 0.473 | 0.507 | 9.376 | 0.513 | 0.560 | 0.569 | 0.585 |
| MotionDiff(AAAI'23) | 15.353 | 0.411 | 0.509 | 0.508 | 0.536 | 5.931 | 0.232 | 0.236 | 0.352 | 0.320 |
| **MOOT(Ours)** | **18.404** | **0.342** | **0.303** | **0.301** | **0.317** | **16.894** | **0.203** | **0.106** | **0.213** | **0.161** |

**Table 2: The comparison results between the proposed method and state-of-the-art methods on ACCAD and HumanAct-12 datasets. The best results are in bold. Lower is better for all metrics except the APD metric.**

| Method | GRAB | | | | | HumanAct-12 | | | | |
|---|---|---|---|---|---|---|---|---|---|---|
| | APD ↑ | ADE ↓ | FDE ↓ | MMADE ↓ | MMFDE ↓ | APD ↑ | ADE ↓ | FDE ↓ | MMADE ↓ | MMFDE ↓ |
| DLow(ECCV'20) | 14.295 | 0.432 | 0.924 | 1.261 | 2.362 | 0.532 | 1.328 | 1.371 | 0.926 | 0.734 |
| MOJO(CVPR'21) | 17.353 | 0.411 | 0.509 | 0.508 | 0.536 | 5.931 | 0.232 | 0.236 | 0.352 | 0.320 |
| MoDi(CVPR'23) | 25.349 | 1.993 | 3.141 | 2.042 | 3.202 | 14.357 | 0.452 | 0.621 | 0.532 | 0.481 |
| HP-GAN(CVPRW'18) | 18.132 | 0.448 | 0.533 | 0.514 | 0.544 | 13.214 | 0.858 | 0.867 | 0.847 | 0.858 |
| GSPS(ICCV'21) | 17.251 | 0.544 | 0.595 | 0.716 | 0.883 | 12.346 | 0.461 | 0.560 | 0.522 | 0.569 |
| MDM(arXiv'22) | 15.622 | 0.311 | 0.504 | 0.408 | 0.516 | 10.156 | 0.419 | 0.541 | 0.516 | 0.572 |
| Actor(ICCV'21) | 11.740 | 0.493 | 0.592 | 0.550 | 0.599 | 13.538 | 0.573 | 0.290 | 0.564 | 0.440 |
| BeLFusion(ICCV'23) | 15.310 | 0.432 | 0.526 | 0.534 | 0.557 | 14.199 | 0.992 | 0.448 | 0.541 | 0.526 |
| **MOOT(Ours)** | **29.404** | **0.204** | **0.326** | **0.513** | **0.487** | **18.496** | **0.331** | **0.251** | **0.212** | **0.241** |

**Table 3: The influence of the introduced OT on the diversity and accuracy results.**

| Method | GRAB | | | | | HumanAct-12 | | | | |
|---|---|---|---|---|---|---|---|---|---|---|
| | APD ↑ | ADE ↓ | FDE ↓ | MMADE ↓ | MMFDE ↓ | APD ↑ | ADE ↓ | FDE ↓ | MMADE ↓ | MMFDE ↓ |
| MoDi(3D Conv+linear layers) | 15.402 | 0.462 | 0.625 | 1.491 | 0.661 | 14.357 | 0.452 | 0.621 | 0.532 | 0.481 |
| MOOT (3D Conv+OT) | 20.865 | 0.291 | 0.348 | 0.526 | 0.541 | 18.096 | 0.411 | 0.293 | 0.322 | 0.301 |
| **MOOT** | **29.404** | **0.204** | **0.326** | **0.513** | **0.487** | **18.496** | **0.331** | **0.251** | **0.212** | **0.241** |

## 6.1 Comparison to Existing Methods

Table 1 and Table 2 summarize the diversity and accuracy results of the proposed method and baseline methods on the used human motion datasets over VAEs, GANs, and DMs-based methods. From the empirical evidence, it is observed that the proposed method consistently outperforms all the baselines based on all the evaluation metrics, particularly demonstrating significant improvements in terms of diversity measures, under the setting of unconditional human motion synthesis. Specifically, MOOT demonstrates a remarkable performance improvement over VAE-based approaches, surpassing these methods by a margin of approximately 8 percentage points in terms of the diversity (APD) metric. Similarly, MOOT achieves approximately 7 percentage points higher scores than GAN-based approaches, indicating its ability to generate more accurate and diverse human motions. Furthermore, concerning the DMs-based approaches, including MDM, Belfusion, and MotionDiff, MOOT achieves approximately 4 percentage points higher scores concerning the diversity metric. Overall, all these results emphasize

that MOOT can significantly improve the diversity while assuring the accuracy of unconditional synthesized human motions.

## 6.2 Ablation Studies

Ablation studies are conducted to justify the influence of introduced optimal matching module in MOOT on diversity and accuracy, as shown in Table 3. Note that MoDi [26] presents a StyleGAN-based style generative model. They utilized 3D convolutions network as the backbone, and employed a mapping network that mapped noise into latent space, which is implemented by the MLP with several DNNs linear layers. To demonstrate the efficacy of the proposed MOOT method, we first utilize the same backbone network as MoDi, and tests different mapping styles to assess their impact on diversity and accuracy metrics. From the results, it is observed that the mapping styles have the greatest impact on the diversity metric of the results. Secondly, we test the difference of different backbone networks for the results, and using GRU and transformer as the encoder and generator network. These results show slightly

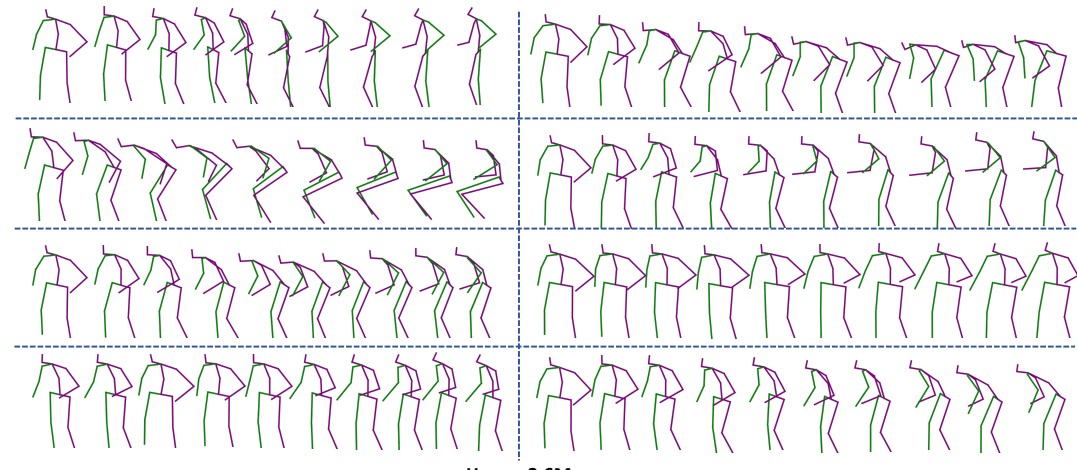

**Human3.6M**

**Figure 5: Qualitative results. The synthesized high-quality, diverse human motion sequences on the Human3.6M dataset.**

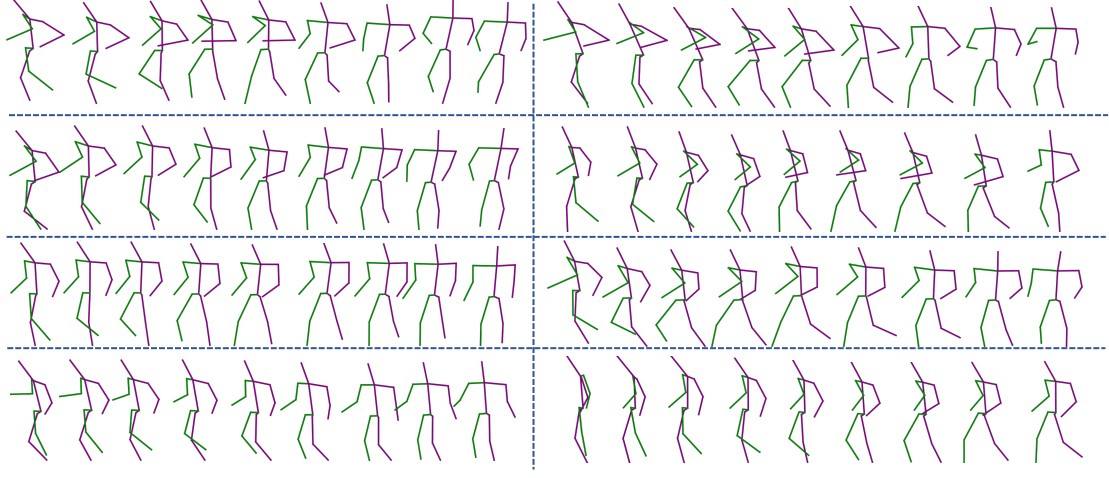

**HumanEva-I**

**Figure 6: Qualitative results. The synthesized multiple human motion sequences on the HumanEva-I dataset.**

improvement on the accuracy metrics. These findings again indicate that the influence of MOOT in enhancing the diversity while assuring the accuracy of unconditional human motion synthesis.

## 6.3 Qualitative Results

In this section, we show the qualitative results of the proposed method on the Human3.6M and HumanEva-I datasets. As depicted in Figure. 5 and Figure. 6. All human motion sequences are unconditionally generated from random noise. Specifically, we first sample a random vector from noise distribution, and map it into a well-structured latent space using the introduced optimal matching, then generate diverse human motions combining the generator. These qualitative results demonstrate that MOOT can generate diverse and coherent human motion sequences.

## 7 CONCLUSION

This paper introduces a novel method called MOOT for unconditional human motion synthesis. MOOT aims to address the issues of mode collapse and mode mixture in generated human motion sequences through the optimal transport mapping. Firstly, the MOOT method employs a human motion reconstruction network to learn the manifold embedding of human motions. This module combines GRU and transformer networks as the encoder and generator backbone network to capture temporal smoothness and spatial relationships among human motions. Secondly, MOOT utilizes the optimal transport mapping to align the noise distribution with the latent space distribution of human motions. The optimal transport mapping is computed using gradient mapping based on Brenier's theorem, which helps alleviate the issue of discontinuity caused by DNNs. Effective exploration of the Brenier potential and optimal transport mapping is achieved through optimization algorithms, thereby overcoming the problems of mode collapse and mode mixture that exist in current unconditional human motion generation methods. Finally, extensive experimental results demonstrate the efficacy of the proposed MOOT method in the task of unconditional human motion synthesis.

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
