# OpenReview forum: "Towards Efficient and Diverse Generative Model for Unconditional Human Motion Synthesis"
_acmmm.org/ACMMM/2024/Conference — MM2024 Poster_

### Official Review · Reviewer_VJbU · 2024-05-23

**Rating:** 4
**Confidence:** 2

**Summary:**

The paper presents a new method for unconditional motion generation. The method is composed of 2 parts. First a GRU and Transformer based VAE to learn to reconstruct motion and obtain a good latent representation of the human motion. the second part is a optimal mapping algorithm to map noise to the VAE latent encoding using the theory of optimal transportation mapping with the goal of generating better motion and avoiding common pitfalls of unconditional generative models (e.g. converging toward a mean pose). The proposed method outperforms the current state of the art by a good margin.

**Strengths:**

The extensive experiments show clear improvements over the state of the art.

The idea of replacing the network used to map noise to latent encoding by a simple algorithm based on the theory of optimal transportation mapping is interesting and could solve common issues of generative model while perhaps reducing complexity.

Extensive preliminaries that help understanding the rest of the paper.

**Limitations:**

For experiments how do the authors deal with the fact that they compare against some conditional method? e.g ACTOR is conditioned on class while MotionDiff is condictioned on past motion. What steps were taken to ensure a fair comparison ?

Regarding quantitative results. How are accuracy metrics (ADE, FDE) computed for the proposed method ? Since it is unconditional how are the generated samples compared to the samples of the ground truth that contains several classes with diverse motions?

Why isn't MotionDiff used in the comparisons on GRAB and HumanAct-12 ? It had the best results on HumanEva-I.

There is no qualitative comparison with the state of the art. Figures 5 and 6 only show samples from the proposed method. It would have been good to be able to compare the quality of the results with other methods. Also videos would have been appreciated to better evaluate the quality of the generated motions.

A study of the complexity of the model or of the inference time would have been interesting as the proposed approach diverge from existing methods and could have given additional argument in favor of the proposed method.

Some insight on the possible applications of unconditional generation would have been interesting.

The Optimal Matching Modeling part could be a bit clearer. Maybe put all longer equations outside the text ?

line 518 "This generator transformed a known continuous distribution, such as Gaussian white noise, into the real distribution of human motions" Is it actually the case? I don't think it output the real distribution but only something resembling it.

line 546 "𝑧𝑖 ∈ R𝑑 , which is a Dirac measure, 𝜈 =..." Is v the dirac measure or is it zi ?

**Suitability:**

3

---

### Official Review · Reviewer_RDLj · 2024-05-23

**Rating:** 2
**Confidence:** 4

**Summary:**

This work presents an alternative way of solving the unconditional human pose generation problem, where the task is to generate a sequence of 3D poses of human motion such that they cover the data distribution, as assumed by a generative modeling objective. The authors propose an approach that is based on semi-discrete Optimal Transport (OT), where instead of having a Neural Network directly approximate the transport map between the prior (noise) distribution and the latent space, an OT map is first calculated based on the Brenier potential to map the prior distribution into the latent space, and then a neural network generates the corresponding motion sequence. In order to embed the motion sequence in latent space, an AutoEncoder based on GRU and Transformer is proposed. Several experiments validate the proposed approach, followed by ablation studies that motivate the design choices.

**Strengths:**

- The experimental results are extremely impressive. MOOT seems to outperform all other human motion generation models, showing that the proposed model does a solid job of capturing the data distribution. This is even more impressive when considering the different type of competitor models that the authors compare with.

- Overall, the paper is very well-written and self-contained. The explanations of the semi-discrete OT problem and how it solves the mapping from the prior distribution to the latent space are clear and help with the presentation of the work.

**Limitations:**

- The biggest weakness of the paper is the novelty, particularly the lack of reference to [1]. Essentially, the proposed model is identical to [1], who proposed using semi-discrete OT in an AE architecture to improve generation and tackle the issue of singularity sets in end-to-end NN-based generative models. Both the problem motivation, solution, algorithm for the calculation of the map (from the Brenier potential), and the extension of the transport map so that sample generation can be performed. Furthermore, the encoder-decoder architecture uses typical NNs used for sequential data (GRU + Transformer) and follows the usual L2 loss to learn the latent space. Thus, the proposed work simply applies these techniques to 3D human poses. A major point of concern is that the use of the semi-discrete OT map for generating data is presented as essentially the main contribution, while it has already been presented in [1].

Minor:
- The critiques of generative models in lines 130-156 are only somewhat true. While VAEs are used with Gaussian priors in practice, this is not a necessary choice. The prior for Variational Inference can be chosen based on the scenario. There is work that also proposes a mixture of priors with learnable components [2]. Furthermore, DDMs are naturally slow under the Markov assumption, which is upheld in the vanilla formulation, but there are several follow-up works that speed up the inference [3]. Nevertheless, the authors compare with prominent generative solutions in their task, so these details would simply make the presentation of the paper more technically correct.

- The metrics used for the evaluation have to be discussed in more detail. In the current version, it is difficult to understand what the results tell us besides that MOOT outperforms all methods. In would be helpful for the reader to understand what MOOT improves the most and understand why, by looking at the gap of a particular evaluation metric.

[1] An, Dongsheng, et al. "AE-OT: A new generative model based on extended semi-discrete optimal transport." ICLR 2020

[2] Tomczak, Jakub, and Max Welling. "VAE with a VampPrior." International conference on artificial intelligence and statistics. PMLR, 2018.

[3] Song, Jiaming, Chenlin Meng, and Stefano Ermon. "Denoising Diffusion Implicit Models." International Conference on Learning Representations. 2020.

**Suitability:**

2

---

### Official Review · Reviewer_3dhe · 2024-05-25

**Rating:** 6
**Confidence:** 4

**Summary:**

This paper is about the problem of unconditional human motion generation from historical observation. The authors focus on an important pain point of this problem "the generated human motions tend to heavily concentrate on densely populated regions of the data distribution", and attribute this problem to the inherent limitation of DNNs in representing only continuous maps. Under this motivation, the authors propose a new method based on optimal transport in the latent space, to transfer the continuous latent distribution to a multimodal discrete distribution. The authors show in the experiments that by doing this, the models can generate diverse motion sequences while keeping the diversity.

**Strengths:**

The work is very well motivated and well presented. It has been observed in the community that the unconditional motion generation models tend to collapse to stationary sequences, especially the poses that appear most frequently in the data set. Motivated from this point, this work present a novel approach to tackle this limitation, with a solid theoretical proof and comprehensive experiment results. I do like this paper. Although I raised many limitations later, this doesn't say a criticism of the article, but I hope these suggestions can make the article better.

**Limitations:**

- As a work on generative AI, it is less convincing to merely show the quantitative results in the table. Instead, I would prefer to see more qualitative demos, i.e. the generated sequences in a video. The images (Fig. 5 and Fig. 6) showing the generated poses, which I understand that they follow the previous work, are not sufficient in today’s research on generative AI.
- Missing reference
  - the theory of optimal transport is not developed in this article, so the Section 3 should be proper referenced, such as [1]. Specifically, the “Kantorovich Relaxation” dates back to the 1940s, it is not a recent development
  - a highly related work [2] should be involved and discussed, which also aims for boost generation quality by optimizing the sampling space
- Typos
  - line 36, Denoising Diffusion Probabilistic Models usually refers to DDPMs, or Diffusion Models, not DMs
- Since this work optimize a OT mapping  to a discrete set (or a multimodal Dirac distribution), I'm curious how this method compares with learning a VQ-VAE model with soft assignment (i.e. the Gumbel-Softmax VQ). A pretrained VQ model can be easily adapted to this task with a auxiliary autoregressive prior model (i.e. GPT), which was shown in a related text-to-motion generation work [3]
- As discussed in [4], the performance on low ADE and hight APD may due to that the model tends to generate one very accurate future sequence (low ADE), and many low quality and unreal sequences (high APD). So it is better to also show other metrics, such as the median value of ADE/FDE in [4] and the FID and action prediction accuracy in [4,5]

[1] Peyré, Gabriel, and Marco Cuturi. ""Computational optimal transport: With applications to data science." Foundations and Trends in Machine Learning, 2019
[2] Dang, Lingwei et al. "Diverse Human Motion Prediction via Gumbel-Softmax Sampling from an Auxiliary Space." ACM MM, 2022
[3] Zhang, Jianrong et al. "T2M-GPT: Generating Human Motion from Textual Descriptions with Discrete Representations." CVPR, 2023
[4] Bie, Xiaoyu et al. "HiT-DVAE: Human Motion Generation via Hierarchical Transformer Dynamical VAE", arXiv, 2022
[5] Petrovich, Mathis, Michael J. Black, and Gül Varol. "Action-conditioned 3d human motion synthesis with transformer vae." ICCV, 2021

**Suitability:**

2

---

### Official Review · Reviewer_EcaH · 2024-06-01

**Rating:** 3
**Confidence:** 4

**Summary:**

The paper presents MOOT, a novel model for synthesizing diverse and high-quality human motion sequences without conditions. It addresses the limitations of existing methods like VAEs, GANs, and DMs in generating diverse motions. MOOT achieves this through a two-stage process, employing an encoder-generator framework combining GRU and Transformer networks for structured latent space learning, and an optimal transport map, extended to a piecewise linear mapping, to align a noise distribution with the learned latent space. This approach effectively enhances diversity in generated motions.

**Strengths:**

Enhanced Diversity and Quality: MOOT is designed specifically to generate human motion sequences that are both diverse and coherent. By aligning a noise distribution with a well-structured latent space of human motions, it ensures the synthesized motions are not only realistic but also span a broad range of possible movements, thus improving upon the limitations of previous methods.

Structural and Temporal Coherence: The employment of a human motion reconstruction network, which integrates GRU and Transformer networks, captures both the temporal continuity and spatial relationships in human motion sequences. This leads to more realistic motion synthesis with natural transitions and spatial plausibility.

Technical Rigor: The paper demonstrates a solid theoretical foundation with the use of Brenier's theorem for computing the optimal transport mapping, which provides a continuous solution, helping to overcome the discontinuity issues associated with Deep Neural Networks (DNNs) in approximating transport maps.

Comprehensive Evaluation: The experimental design includes rigorous testing on multiple benchmark datasets such as Human3.6M, HumanEva-I, HumanAct12, and GRAB. The study encompasses a range of parameters, evaluation metrics, and baseline comparisons, validating MOOT's superior performance in generating diverse human motion sequences.

Practical Relevance: The ability to generate diverse and natural human motion sequences has direct implications for several application areas, including animation, robotics, autonomous systems, and human-computer interaction, highlighting the practical significance of the research.

**Limitations:**

1. This article is a lot like human motion prediction, where a lot of the experimental setups, and comparison methods are derived from motion prediction. But the title of the author's paper is motion generation. Please explain and clarify thes.
2.Use the last frame of the observation sequence for motion synthesis, why not use more frames? Why use only the last frame? Can we not use this frame?
3.I think that the author did not explain the differentiation between the two tasks of motion synthesis as well as motion prediction, and there is a logical confusion in the article.
4. There are many grammatical problems, line 100, incomplete in the sentence, line 204, the word 'describe' should be' describes',and so on.
5. The explanation is not clear in Figure 1. What do the different colored dots in Figure 1 mean? What do the lines indicate? It is suggested that the author should elaborate in CAPTION.

**Suitability:**

2

---

### Meta-Review · Area_Chair_66bM · 2024-07-02

**Recommendation:** Accept (Poster)
**Confidence:** 3

**Metareview:**

The four reviewers' decisions are split (A, BA, BR, WR). The method using optimal transport is novel in this area, and its effectiveness has been verified by experiments. However, the concern is that optimal transport has been used in other areas, and the evaluation metric is rather biased. But still, I lean toward accepting this paper.